# Scoping review of molecular biomarkers associated with fatigue, stress, and depression in stroke survivors: A protocol

**Tarynn Potter**[1]*, **Nisal Gange**[2], **Eliza Whiteside**[1], **Prajwal Gyawali**[1]

**1** School of Health and Medical Science, Faculty of Health, Engineering and Sciences, University of Southern Queensland, Toowoomba, Queensland, Australia, **2** Geriatric Adult Rehabilitation and Stroke Service (GARSS), Darling Downs Health Service (DDHS), Toowoomba Hospital, Toowoomba, Queensland, Australia

* tarynn.potter@usq.edu.au










**Data Availability Statement:** No datasets were generated or analysed during the current study. All relevant data from this study will be made available upon study completion.

## Abstract

The prevalence of stroke increases each year and while mortality from stroke has decreased, the prevalence of comorbidities such as anxiety, depression and fatigue affects as many as 75% of stroke survivors. The aetiology of post-stroke fatigue is not clear, although it has been shown to be interrelated with comorbidities such as stress and depression. Due to the interconnected nature of these comorbidities, it is important to improve the specificity of diagnosis and identify novel therapeutic targets to improve the quality of life for stroke survivors. The investigation of molecular biomarkers associated with post-stroke stress, fatigue, and depression may shed light on the relationships between comorbidities and also contribute to the development of novel diagnostics and therapies. Several biomarkers have been identified for stress, depression, and fatigue, some of which are specific to stroke survivors. However, there remain several gaps in understanding, particularly in relation to the physiological mechanisms underlying these side effects and molecular biomarkers associated with post-stroke fatigue. The aim of this scoping review protocol is to outline the methodologies that will be used to provide a comprehensive understanding of the current literature on biomarkers associated with post-stroke fatigue, stress, and depression, informing future research questions.

## Introduction

Stroke is one of the five leading causes of death in Australia [1], and the burden resulting from stroke has increased globally [2]. While more people survive stroke due to medical interventions [3], between 25%-74% of stroke survivors experience ongoing psychological comorbidities [4–7]. These comorbidities include symptoms of stress in 68% of patients, anxiety in 52.9% of patients, and depression in 74.5% of patients [6, 8]. These psychological comorbidities can significantly affect quality of life and their impacts differ among patients dependent on demographic and mental health status [6].

**Funding:** This research has been supported by an Australian Government Research Training Program Scholarship. The funders had no role in study design, data collection and analysis, decision to publish, or preparation of the manuscript.

**Competing interests:** The authors have declared that no competing interests exist.

Post-stroke fatigue is reported by patients to be among the worst symptoms to cope with and is reported by up to 75% of stroke survivors [9]. Higher levels of fatigue have been associated with symptoms of depression [9] and reduced sleep quantity has been associated with the development of post-stroke stress and anxiety [8]. Stress has been shown to significantly affect stroke recovery, correlating increased stress levels to higher functional impairment [10], is negatively associated with health outcomes for stroke survivors [10–12], and often precedes depression and anxiety [13]. Despite the high prevalence of fatigue in stroke survivors, a lack of accurate assessment for post-stroke fatigue inhibits the development of more effective clinical interventions [14]. Quality of life for stroke survivors is negatively affected by fatigue, stress, and depression [8, 15] however, measurement and treatment of these comorbidities is inconsistent [11, 16]. Treatment for post-stroke fatigue, anxiety and depression would provide stroke survivors with improved quality of life and health outcomes.

The relationship between post-stroke fatigue, stress and depression is complex. Although shown to be interrelated, the relationship between fatigue and side effects such as stress and depression remains to be clarified [9, 11]. These comorbidities may be part of a perpetual cycle, compounding negative outcomes for stroke survivors [8, 14]. The complexity of these relationships carries through to treatment, and as a result there is currently no best practice [17]. Treatment post-stroke fatigue generally relies on individual clinician recommendation based on personal experience rather than evidence-based treatment [16]. In addition to inconsistencies in treatment, the treatment options currently available for stroke survivors are limited in their efficacy [18].

A thorough understanding of the biomarkers associated with post-stroke symptoms, could provide biological measurement processes for follow up or early diagnosis. Limited studies have undertaken molecular biomarker analysis for post-stroke fatigue, although several have been conducted on the molecular biomarkers associated with chronic fatigue syndrome [19–21]. Furthermore, several serum and salivary biomarkers have been identified in stroke survivors experiencing stress and depression [22, 23]. This leaves a significant gap in the understanding of molecular biomarkers associated with post-stroke fatigue. There is, however, considerable overlap in the identified biomarkers for stress, depression, and fatigue, which provides an opportunity to elucidate the interrelated biology of these conditions in stroke survivors. By determining the biomarkers associated with post-stroke stress, fatigue, and depression, higher specificity of diagnosis can potentially be achieved, and therapeutic targets can be developed. Although the benefit of elucidating these biomarkers in stroke survivors has potential for novel screening and therapeutic interventions, this is the first review that will map potential biomarkers specific to post-stroke fatigue.

Overall, the number of molecular biomarkers associated with stress, fatigue, and depression are quite significant and provide various targets for further investigation due to the complex mechanism of action behind each of these conditions. In addition, the current notion that affective disorders, including depression, have a common physiological mechanism within a clinical spectrum of severity [24] highlights the necessity for greater understanding of molecular biomarkers to provide insight into these mechanisms. Unfortunately, this has yet to translate into effective molecular diagnostic tools, particularly in stroke survivors. While some research has been conducted into biomarkers associated with stress and depression in stroke survivors, there is a lack of literature available on molecular biomarkers associated with post-stroke fatigue. Due to this lack of literature, the potential for uncovering novel diagnostic and therapeutic biomarkers is high. In addition, the complexity of the interrelationship between stress, fatigue, and depression presents an opportunity to build on the current baseline understanding of the biomarkers associated with these comorbidities and allow more robust comparative analyses moving forward.

## Aims and objectives

This scoping review will provide a review of current literature and aims to:

- Identify molecular biomarkers associated with post-stroke fatigue, stress, and depression
- Describe the relationships between post-stroke fatigue, stress, and depression
- Identify the approaches that have been used to measure molecular biomarkers in patients with post-stroke fatigue, stress, and depression
- Inform future research to identify novel biomarkers for diagnosis and therapeutic intervention for stroke survivors
- Identify questions for future research into molecular biomarkers for post-stroke fatigue, stress, and depression as independently and concurrently occurring morbidities

## Methods and analysis

This review will be conducted in accordance with a scoping review methodology using the framework first outlined by Arksey and O'Malley [25] and in line with guidance from Joanna Briggs Institute [26]. This method was chosen to enable the effective mapping and summarising of a broad range of existing research outcomes, to examine the methods used when conducting biomarker measurements and to identify gaps in collective understanding in line with the previously outlined aims and objectives.

The framework that will be used involves refining the research question; identifying the relevant research publications that have addressed the research question; publication selection; charting the data; and collating, summarising and reporting the results [25]. The PRISMA Extension for Scoping Reviews [27] checklist will be used for reporting. Studies will be selected according to the eligibility criteria in Table 1. Eligibility criteria are based on the PICO (participant, intervention, comparator, and outcome) format.

### Stage 1

**Refining the research question.** Initially we endeavoured to review the literature primarily related to molecular biomarkers associated with post-stroke fatigue. It was identified that

**Table 1. Eligibility criteria for screened references.**

| | Inclusion | Exclusion |
|---|---|---|
| Population | Stroke or transient ischemic attack survivors | Non-human subject |
| | Any age | |
| Interventions | Molecular biomarkers | Oxidative stress biomarkers not discussed in association with stress, fatigue, or depression |
| Comparator | Not applicable | Not applicable |
| Outcomes | Altered biomarker profile | No exclusion criteria |
| Publication | Peer-reviewed journal articles of any design–qualitative or quantitative. | Reviews |
| | | Commentaries/opinion papers |
| | | Letters |
| | | Meta-analyses |
| | | Not published in English |
| | | Unpublished studies |


there was insufficient independent research into this topic to answer this question with a systematic review. Due to the interconnected nature of post-stroke fatigue with stress and depression, we instead endeavoured to identify molecular biomarkers that are associated with each of these comorbidities. By identifying the molecular biomarkers associated with post-stroke fatigue, stress, and depression we can potentially elucidate the interrelated biology of these comorbidities. The aims and objectives section of this protocol outlines the research questions.

## Stage 2

**Identifying relevant studies.** In consultation with a Senior Research Librarian, the initial search strategy was developed, as shown in Table 2. The strategy drew on common terms for the key themes being investigate. The terms searched included 'post-stroke', 'poststroke', 'post stroke', 'biomarker', 'fatigue', 'stress' and 'depression'. A preliminary search was conducted to refine search strategy which included the addition of terms 'stroke recovery' and 'stroke rehabilitation'. The search strategy was developed in Scopus and expanded to PubMed, Web of Science, Science Direct and EBSCOHost. No publication type or date limits were applied.

## Stage 3

**Study selection.** Studies identified in the search will be collated in EndNote X9.3.1 and exported to the Joanna Briggs Institute (JBI) SUMARI for screening of titles and abstracts. Each will be independently reviewed by two authors in line with the proposed criteria outlined in Table 1. Conflicts will be resolved in consultation with an independent third party. If no consensus is reached, the article will progress for further review. Following initial screening, full text articles will be independently reviewed by two authors with any conflicts resolved by consensus between the authors. A PRISMA flow diagram will be used to report the selection process [28].

While not required as part of the scoping review methodology, all articles will be critically appraised. This is to allow assessment of the efficacy of biomarkers identified by analysing the quality of methods used for extraction and measurement of these markers. Critical appraisal will be undertaken in JBI SUMARI using the Cochrane Risk of Bias Quality Appraisal Tool [29] for quantitative studies, and the JBI SUMARI Checklist for Qualitative Research [30]. Two independent authors will complete the critical appraisal and discuss and resolve any discrepancies.

## Stage 4

**Charting the data.** The data will be charted to comprehensively capture the information from studies that meet the eligibility criteria as outlined in the JBI Manual for Evidence Synthesis [31]. This will include capturing the participant and sample size, clinical details pertinent to stroke, study methods and key findings relevant to our aims and objectives. In addition, we will record the nature in which biomarkers were studied for monitoring, diagnosis, measurement, prediction or to elucidate the relationship to biological processes. A standardised electronic form in Excel will be utilised with the headings outlined in Table 3, and each article will be detailed by one author and verified by the second author. Resolution of discrepancies will be resolved via consensus between the authors. Consistency will be maintained throughout the process by consultation between authors and any considerable alterations to the charted data will be recorded.

## Stage 5

**Collating, summarising, and reporting results.** The results of the articles charted will be summarised with a focus on the research questions outline in the aims and objectives of this

**Table 2. Search strategy development.**

| Key Words: "post-stroke" OR poststroke OR "stroke recovery" OR "stroke rehabilitation" OR "post stroke" AND biomarker AND fatigue? OR stress OR depression | | | | |
|---|---|---|---|---|
| **Search Strategy** | **Database** | **n results** | **Field Search** | **Limits/Filters** |
| ("post-stroke" OR poststroke OR "stroke recovery" OR "stroke rehabilitation" OR "post stroke") AND biomarker? AND (fatigue? OR stress OR depression) | Scopus | 123 | Title-Abstract-Keyword | None |
| ("post-stroke" OR poststroke OR "stroke recovery" OR "stroke rehabilitation" OR "post stroke") AND biomarker? AND (fatigue? OR stress OR depression) | Scopus | 120 | Title-Abstract-Keyword | English Language only |
| ("post-stroke" OR poststroke OR "stroke recovery" OR "stroke rehabilitation" OR "post stroke") AND biomarker? AND (fatigue? OR stress OR depression) AND NOT oxidative) | Scopus | 79 | Title-Abstract-Keyword | English Language only |
| ("post-stroke" OR poststroke OR "stroke recovery" OR "stroke rehabilitation" OR "post stroke") AND biomarker? AND fatigue? AND NOT oxidative) | Scopus | 0 | Title-Abstract-Keyword | English Language only |
| ("post-stroke" OR poststroke OR "stroke recovery" OR "stroke rehabilitation" OR "post stroke") AND biomarker? AND fatigue?) | Scopus | 0 | Title-Abstract-Keyword | English Language only |
| ("post-stroke" OR poststroke OR (stroke W/3 recovery) OR (stroke W3 rehabilitation) OR "post stroke") AND biomarker? AND (fatigue? OR stress OR depression) AND NOT oxidative) | Scopus | 81 | Title-Abstract-Keyword | English Language only |
| stroke? AND biomarker? AND fatigue? | Scopus | 0 | Title-Abstract-Keyword | None |
| stroke? AND biomarker? AND fatigue? | PubMed | 55 | All fields | None |
| ("post-stroke" OR poststroke OR (stroke W/3 recovery) OR (stroke W3 rehabilitation) OR "post stroke") AND biomarker? AND (fatigue? OR stress OR depression) AND NOT oxidative) | PubMed | 30 | All fields | None |
| ("post-stroke" OR poststroke OR (stroke W/3 recovery) OR (stroke W3 rehabilitation) OR "post stroke") AND biomarker? AND (fatigue? OR stress OR depression) | PubMed | 144 | All fields | None |
| ("post-stroke" OR poststroke OR "stroke recovery" OR "stroke rehabilitation" OR "post stroke") AND biomarker? AND (fatigue? OR stress OR depression) | PubMed | 158 | All fields | None |
| ((((((ALL = (("post-stroke")) OR ALL = (poststroke)) OR ALL = ("stroke recovery")) OR ALL = ("stroke rehabilitation")) OR ALL = ("post stroke"))) AND ALL = (biomarker*)) AND ALL = ((fatigue* OR stress or depression)) | Web of Science | 144 | All fields | None |
| ALL = ("post-stroke" OR poststroke OR "stroke NEAR/3 recovery" OR "stroke NEAR/3 rehabilitation" OR "post stroke") AND ALL = (biomarker*) AND ALL = (fatigue* OR stress or depression) NOT ALL = (oxidative) | Web of Science | 94 | All fields | None |
| post-stroke OR poststroke OR stroke recovery OR stroke rehabilitation OR "post stroke" | EBSCOHost: Academic Search Ultimate, APA PsycArticles, APA PsycInfo, CINAHL with Full Text, Health Source: Nursing/Academic Edition, Psychology and Behavioral Sciences Collection | 57583 | All fields | None |
| (post-stroke OR poststroke OR stroke recovery OR stroke rehabilitation OR "post stroke") AND biomarker* | EBSCOHost: Academic Search Ultimate, APA PsycArticles, APA PsycInfo, CINAHL with Full Text, Health Source: Nursing/Academic Edition, Psychology and Behavioral Sciences Collection | 753 | All fields | None |
| (post-stroke OR poststroke OR stroke recovery OR stroke rehabilitation OR "post stroke") AND biomarker* AND fatigue* | EBSCOHost: Academic Search Ultimate, APA PsycArticles, APA PsycInfo, CINAHL with Full Text, Health Source: Nursing/Academic Edition, Psychology and Behavioral Sciences Collection | 5 | All fields | None |

**Table 2.** (Continued)

| Key Words: "post-stroke" OR poststroke OR "stroke recovery" OR "stroke rehabilitation" OR "post stroke" AND biomarker AND fatigue? OR stress OR depression | | | | |
|---|---|---|---|---|
| **Search Strategy** | **Database** | **n results** | **Field Search** | **Limits/Filters** |
| (post-stroke OR poststroke OR stroke recovery OR stroke rehabilitation OR "post stroke") AND biomarker* AND (fatigue* OR stress OR depression) | EBSCOHost: Academic Search Ultimate, APA PsycArticles, APA PsycInfo, CINAHL with Full Text, Health Source: Nursing/Academic Edition, Psychology and Behavioral Sciences Collection | 177 | All fields | None |
| [abstract] (fatigue OR depression OR stress) AND (post-stroke OR poststroke OR "stroke recovery" OR "stroke rehabilitation" OR "post stroke") AND [all fields] biomarker (all fields) | Science Direct | 97 | Abstract-All fields | None |
| ((post-stroke OR poststroke OR "stroke recovery" OR "stroke rehabilitation" OR "post stroke") AND biomarker* AND (fatigue* OR stress OR depression)) | Scopus | 172 | Title-Abstract-Keywords | None |
| (post-stroke OR poststroke OR "stroke recovery" OR "stroke rehabilitation" OR "post stroke") AND (fatigue? OR stress OR depression) ALL Fields biomarker? | Science Direct | 139 | Title-Abstract-Keyword | None |
| ("post-stroke" OR poststroke OR "stroke recovery" OR "stroke rehabilitation" OR "post stroke") AND biomarker? AND (fatigue? OR stress OR depression) | PubMed | 171 | All fields | None |
| ("post-stroke" OR poststroke OR "stroke recovery" OR "stroke rehabilitation" OR "post stroke") AND biomarker? AND (fatigue? OR stress OR depression) | Web of Science | 98 | All fields | None |
| ("post-stroke" OR poststroke OR "stroke recovery" OR "stroke rehabilitation" OR "post stroke") AND biomarker? AND (fatigue? OR stress OR depression) | EBSCOHost: Academic Search Ultimate, APA PsycArticles, APA PsycInfo, CINAHL with Full Text, Health Source: Nursing/Academic Edition, Psychology and Behavioral Sciences Collection | 184 (110*) | All fields | None |
| *n without duplicates | | | | |


**Table 3. Data extraction table headings.**

| Column Title | Summary |
|---|---|
| Author/Study | Author name or Study title |
| Aims/theory/hypothesis/questions | Correlation or association of stress/fatigue/depression with inflammation. Could inflammation predict fatigue or fatigue severity. |
| Nature of study and number of participants | i.e., cross-sectional, prospective, RCT and number of patients and/or controls. |
| Participant details | Including time post-stroke, frequency of collect and when blood collection occurred (within 72 hours post diagnosis, after 3 months post diagnosis or other).<br>Was there a control group (comparative studies). |
| Stroke related clinical information | Including stroke severity (NIHSS, mRS); type (ischaemic or haemorrhagic); left or right side; initial or recurrent stroke. |
| What aspects of stroke recovery discussed | Stress and/or fatigue and/or depression. How were these variables measured and what tools were used? |
| Biomarkers | What biomarkers measured: inflammation, oxidative stress, tryptophan/kynurenine pathway, hormones, genetics (gene, loci, allele, RNA etc). |
| Key findings | Inflammation correlated with fatigue or inflammation predicts fatigue or inflammation caused fatigue or something that authors concluded |
| Pathophysiological basis | why did fatigue correlate with inflammation or why was depression predicted by oxidative stress |
| Other questions related to paper quality assessment | |


protocol. Quantitative data will be tabulated by biomarker type. Qualitative data will be narrated or tabulated by theme. Further description of all results will be provided for context in relation to the research question and any gaps in the literature will be narrated. Interpretation of the results will be conducted within the constraints of the review limitations. Data underlying the findings will be available with the review as supplementary tables. Relevant literature may be missed as grey literature will be excluded. This was intended to ensure assessment of the quality of research included.

## Discussion

As the prevalence of stroke continues to increase and up to 75% of post-stroke patients suffer from anxiety, depression, or fatigue, it is imperative that novel targets are identified for diagnostic and therapeutic agents to improve quality of life for these patients. Previous studies have identified molecular biomarkers that can assist in analysing post-stroke stress and depression, however this is the first review that will map potential biomarkers for post-stroke fatigue. One of the strengths of this study is that biomarkers will be assessed in relation to stress, depression, and fatigue, which will better elucidate the relationship and common mechanisms of these comorbidities in stroke survivors. We anticipate this review will identify gaps in current understanding and identify priorities for future research into molecular biomarkers associated with post-stroke fatigue, stress, and depression. This review may also have further implications on understanding stress, depression, and fatigue outside of stroke care, where the effects on biomarkers are likely to be the same and mechanisms may be similar. The outcome of this review will be relevant to researchers, clinicians, and policy makers.

## Supporting information

**S1 Checklist. PRISMA-P (Preferred Reporting Items for Systematic review and Meta-Analysis Protocols) 2015 checklist: Recommended items to address in a systematic review protocol*.**
(DOC)

**S1 File.**
(PDF)

## Author Contributions

**Conceptualization:** Tarynn Potter, Eliza Whiteside, Prajwal Gyawali.

**Data curation:** Tarynn Potter, Prajwal Gyawali.

**Formal analysis:** Tarynn Potter, Prajwal Gyawali.

**Funding acquisition:** Prajwal Gyawali.

**Investigation:** Tarynn Potter.

**Methodology:** Tarynn Potter, Prajwal Gyawali.

**Project administration:** Tarynn Potter, Eliza Whiteside, Prajwal Gyawali.

**Resources:** Tarynn Potter, Eliza Whiteside, Prajwal Gyawali.

**Supervision:** Nisal Gange, Eliza Whiteside, Prajwal Gyawali.

**Validation:** Tarynn Potter, Nisal Gange, Eliza Whiteside, Prajwal Gyawali.

**Visualization:** Tarynn Potter.

**Writing – original draft:** Tarynn Potter.

**Writing – review & editing:** Tarynn Potter, Nisal Gange, Eliza Whiteside, Prajwal Gyawali.

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
