## [Decision Letter · Decision Letter 0]

17 Oct 2022

PONE-D-22-24829A systematic scoping review of molecular biomarkers associated with fatigue, stress, and depression in stroke survivors: a protocolPLOS ONE

Dear Dr. Potter,

Thank you for submitting your manuscript to PLOS ONE. After careful consideration, we feel that it has merit but does not fully meet PLOS ONE’s publication criteria as it currently stands. Therefore, we invite you to submit a revised version of the manuscript that addresses the points raised during the review process.

We look forward to receiving your revised manuscript.

Kind regards,

Abiodun E. Akinwuntan, PhD, MPH, MBA

Academic Editor

PLOS ONE

Journal Requirements:

Reviewers' comments:

Reviewer's Responses to Questions

**Comments to the Author**

1. Does the manuscript provide a valid rationale for the proposed study, with clearly identified and justified research questions?

Reviewer #1: Yes

Reviewer #2: Yes

2. Is the protocol technically sound and planned in a manner that will lead to a meaningful outcome and allow testing the stated hypotheses?

Reviewer #1: Yes

Reviewer #2: Yes

3. Is the methodology feasible and described in sufficient detail to allow the work to be replicable?

Reviewer #1: Yes

Reviewer #2: Yes

4. Have the authors described where all data underlying the findings will be made available when the study is complete?

Reviewer #1: Yes

Reviewer #2: Yes

5. Is the manuscript presented in an intelligible fashion and written in standard English?

Reviewer #1: Yes

Reviewer #2: Yes

6. Review Comments to the Author

You may also provide optional suggestions and comments to authors that they might find helpful in planning their study.

Reviewer #1: Some major and minor points to raise regarding this study protocol.

Specific comments:

1. Suggest to call this a "scoping review" rather than a "systematic scoping review".

2. Please change "Unfortunately, this is yet to translate" to "Unfortunately, this has yet to translate".

3. There was a previous systematic review on the same topic published in 2016 (citation: pubmed.ncbi.nlm.nih.gov/26891661) and a few other reviews since. What therefore is the rationale for the present review? The prior studies were not referenced by the authors; they should be referenced.

4. The current thinking is that depression likely runs an entire clinical spectrum from mild to severe; there are genetic and neurobiological studies lending support to the notion that these conditions are not discrete categories but rather, have common biological underpinnings and may form at least part of a continuum or affective disorder spectrum (citation: pubmed.ncbi.nlm.nih.gov/32557983). This should be at least briefly mentioned as it has important implications for diagnosis and research and would further strengthen the need for precise biomarkers.

Reviewer #2 (who is also the Academic Editor Abiodun Akinwuntan): Authors are to be commended for this version of the manuscript. However, there are major concerns that need to be addressed before the manuscript is suitable for publication.

A major problem that repeats across the entire manuscript is the inconsistency in and interchangeable use of major descriptors of the protocol including fatigue, anxiety, stress, depression, sleeplessness, and quality of life.

Abstract: Line 23: What these several gaps that remain?

Introduction: Lines 45-62: Too many repetitions of facts. These paragraphs can be shortened into in short and succinct paragraph.

Line 93: Consider replacing "thorough understanding" with "review".

Methods and Analysis: Lines 125-126: How will the understanding lead to increase in specific diagnostic and therapeutic interventions to guide future research? This sentence is confusing and misleading.

Other aspects of the manuscript seem well-written

7. PLOS authors have the option to publish the peer review history of their article (what does this mean?). If published, this will include your full peer review and any attached files.

Reviewer #1: No

Reviewer #2: No

---

## [Author Response · Author response to Decision Letter 0]

6 Dec 2022

Reviewer #1 Feedback:

1. Suggest calling this a “scoping review” rather than a “systematic scoping review”. Title has been adjusted from “systematic scoping review” to “scoping review”. 

2. Please change “Unfortunately, this is yet to translate to” to “ Unfortunately, this has yet to translate”. The sentence on line 103 has been corrected.

3. Previous systematic review on the same topic published in 2016 and a few other reviews since then. What therefore is the rationale for the present review? The prior studies were not referenced by the authors. Lines 75-77 have been updated to better reflect the previous reviews that have been conducted into biomarkers for post-stroke depression. We have also provided a more robust rationale with edits to lines 88-90 to better clarify the knowledge gap the review aims to address. 

4. The current thinking is that depression likely runs an entire clinical spectrum from mild to severe; there are genetic and neurobiological studies lending support to the notion that these conditions are not discrete categories but rather have common biological underpinnings and may form at least part of a continuum or affective disorder spectrum. This should at least be briefly mentioned as it has important implications for diagnosis and research and would further strengthen the need for precise biomarkers. Thank you for flagging this for consideration. This reference has been included and the implications to diagnostic and therapeutic processes outlined on lines 100-103. 

Reviewer #2 feedback:

1. A major problem that repeats across the entire manuscript is the inconsistency in and interchangeable use of major descriptors of the protocol including fatigue, anxiety, stress, depression, sleeplessness, and quality of life. Thank you for flagging the inconsistencies with these descriptors. The protocol has been reviewed and the major descriptors updated to reduce these inconsistencies. This includes updates to line 16 to provide more consistency as well as consideration for clear articulation of the main variables considered in this protocol in the substantial edits for lines 49-61. 

2. Abstract: Line 23: What these several gaps that remain? Lines 22-24 have been updated to include detail of the current gaps in understanding.

3. Introduction: Lines 45-62: Too many repetitions of facts. These paragraphs can be shortened into in short and succinct paragraph. This section has been updated from lines 47-61 to concisely articulate key concepts with greater clarity. 

4. Line 93: Consider replacing "thorough understanding" with "review". Line 112 has been updated with this change. 

5. Methods and Analysis: Lines 125-126: How will the understanding lead to increase in specific diagnostic and therapeutic interventions to guide future research? This sentence is confusing and misleading. This statement has been removed (lines 144-146) as it did not add value to the rationale of this paragraph.

---

## [Decision Letter · Decision Letter 1]

19 Jan 2023

Scoping review of molecular biomarkers associated with fatigue, stress, and depression in stroke survivors: a protocol

PONE-D-22-24829R1

Dear Dr. Potter,

We’re pleased to inform you that your manuscript has been judged scientifically suitable for publication and will be formally accepted for publication once it meets all outstanding technical requirements.

Kind regards,

Abiodun E. Akinwuntan, PhD, MPH, MBA

Academic Editor

PLOS ONE

Additional Editor Comments (optional):

Reviewers' comments:

Reviewer's Responses to Questions

**Comments to the Author**

1. Does the manuscript provide a valid rationale for the proposed study, with clearly identified and justified research questions?

Reviewer #1: Yes

Reviewer #2: Yes

2. Is the protocol technically sound and planned in a manner that will lead to a meaningful outcome and allow testing the stated hypotheses?

Reviewer #1: Partly

Reviewer #2: Yes

3. Is the methodology feasible and described in sufficient detail to allow the work to be replicable?

Reviewer #1: Yes

Reviewer #2: Yes

4. Have the authors described where all data underlying the findings will be made available when the study is complete?

Reviewer #1: No

Reviewer #2: Yes

5. Is the manuscript presented in an intelligible fashion and written in standard English?

Reviewer #1: Yes

Reviewer #2: Yes

6. Review Comments to the Author

You may also provide optional suggestions and comments to authors that they might find helpful in planning their study.

Reviewer #1: Thank you for the revisions. Authors should also describe where all data underlying the findings will be made available when the study is complete.

Reviewer #2: The authors have done a good job responding to each of the comments. I have no additional comments at this time.

7. PLOS authors have the option to publish the peer review history of their article (what does this mean?). If published, this will include your full peer review and any attached files.

Reviewer #1: No

Reviewer #2: No

---

## [Editor Report · Acceptance letter]

24 Jan 2023

PONE-D-22-24829R1 

Scoping review of molecular biomarkers associated with fatigue, stress, and depression in stroke survivors: a protocol 

Dear Dr. Potter:

I'm pleased to inform you that your manuscript has been deemed suitable for publication in PLOS ONE. Congratulations! Your manuscript is now with our production department. 

Kind regards, 

on behalf of

Dr. Abiodun E. Akinwuntan 

Academic Editor

PLOS ONE